# Effects of Age and Popularity of Sport on Differences among Wrestlers’ Parental Support: An Exploratory Study

**DOI:** 10.3390/jfmk8020065

**Published:** 2023-05-15

**Authors:** Ivica Biletic, Hrvoje Karnincic, Mario Baic

**Affiliations:** 1Police Academy—The First Croatian Police Officer, University of Applied Sciences in Criminal Investigation and Public Security, 10000 Zagreb, Croatia; biletic@net.hr; 2Faculty of Kinesiology, University of Split, 21000 Split, Croatia; 3Faculty of Kinesiology, University of Zagreb, 10000 Zagreb, Croatia; mario.baic@kif.unizg.hr

**Keywords:** age categories, parental involvement, combat sport

## Abstract

No research was previously performed on wrestling related to parental support. It is not known whether there are differences in support between younger and older children. The popularity of a sport can be reflected in parental support, and parents may be more inclined towards popular sports. The aim of this research was to examine differences in parental support among wrestlers of different age categories and between those coming from communities in which wrestling is a popular sport versus communities in which it is less popular. The sample of participants consisted of 172 wrestlers. The Parental Support Scale for Children in Sports was applied. Parental willingness to set an example was lower. As far as age is concerned, the period of entry into specialisation is sensitive. At this age, children perceive less parental support (*p* = 0.04) and lower parental belief in the benefits of sports (*p* = 0.01). The popularity of the sport is related to parental support. In environments in which wrestling is popular, parents know the sport better and can participate; therefore, children perceive more parental support. The findings of this study may help coaches to better understand athlete–parent relationships.

## 1. Introduction

Parental support is an important factor in children’s adherence to participation in sports [1], and it can also be a reason for practicing sports [2]. One of the most important socialising forces for children’s entry into sports is the family, especially parents [3,4]. Parental beliefs are often related to their children’s understanding of sport [5,6,7]. Woolgar and Power (1993) were the first to define three basic forms of parental involvement in children’s sport: emotional support, the provision of information, and concrete help [8]. Bosnar (2003) identified two other important aspects: setting an example for children and positive reinforcement, which are related to expectations for children’s sport performances. Expectations should be realistic (neither too low nor too high) to avoid having a negative effect on children’s motivation [9]. Parental support for children in sports was researched through many concepts in numerous studies. Most generally speaking, all these studies and their findings can be reduced to a twofold concept: parental support and parental pressure. In the present research, the topic is parental support. The analysis of parental support can reveal some causal mechanisms behind children either continuing or quitting practicing sports. A better understanding of these mechanisms can increase the number of children engaged in sports. In an attempt to better investigate this topic, a series of questionnaires was created: the Parental Support Scale [10,11], Parental Involvement in Sport Questionnaire (PISQ) [12], Parent-Initiated Motivational Climate Questionnaire [13], and others. All these questionnaires have different foci; therefore, the findings or results of the studies that used them cannot be compared. What are comparable are general conclusions. The Croatian version of the questionnaire was designed by Bosnar and associates in 2003, and it was validated in samples of primary and secondary school children, as well as in samples of children involved in combat sports [9,14,15,16].

Giving up further involvement in sports is negatively correlated with the age of the athlete [17]. It is well known that growing up brings more responsibilities and less free time for training. The question arises as to whether parental support is one of the factors that causes older children to withdraw from sports. The popularity of a sport can be both positively and negatively reflected in parental support. Children’s success in sports can change their parents’ social status [18]. One of the reasons why children play sports is because they are popular, and children who play sports are more popular [18]. It is logical that this phenomenon is more pronounced in popular sports, in which the profits are higher. This can lead to negative consequences, such as an excessive focus on results. Fortunately, such parents are in the minority [19]. Problems with illegal substances in young athletes occur in popular sports for this reason; however, this is not yet a mass phenomenon [20]. Research showed that the most important thing for a man’s popularity is sport, while for women, the most important thing is appearance [21]. In this paper, the focus is on the popularity of a sport in a particular setting. The popularity of a sport may vary in environments that are culturally and environmentally close to each other. For example, football, which is most popular in Europe, is not the most popular sport in the United States (where it is behind basketball, baseball, and American football) [22]. Parents who set overly ambitious goals and participate too much in their children’s sports careers have a larger problem if the sport is popular, and especially if the earnings are high [23]. In previous research, it was demonstrated that parents of higher-standard athletes provide them with more autonomy in their upbringing, whereas the upbringing of lower-standard athletes is characterised by the excessive involvement of their parents [24].

According to the list of the Croatian Wrestling Federation, there are 30 active wrestling clubs in Croatia. In its capital city, Zagreb, and close surrounding areas, there are 14 wrestling clubs (nine in the city and five in the surrounding areas), together with numerous sections in schools. The rest of the 16 clubs are active in 14 Croatian cities. Only two cities recently established second wrestling clubs. Research showed that, in Australia, more children participate in sports where sports facilities are most accessible [25]. This is also the case in Croatia because the number of sports facilities for wrestling in the capital is much greater. However, in the same study, it was found that there were fewer sports facilities and less interest in sports in the capital [25]. In this study, the situation is reversed: half of all clubs and wrestlers in the country are located in the capital. We can assume that this difference is due to the different sizes of the countries. In the Croatian Premier First Wrestling League, six clubs are competing, of which four are from Zagreb. We can feasibly say that one-half of the Croatian wrestling activities are related to the capital city and its surroundings. Moreover, the longest tradition of wrestling is also related to the capital city—as many as six of the first wrestling clubs in Croatia were founded in Zagreb [26], and almost all of the important national and international competitions take place there. Therefore, we can say that, in the capital of Croatia, a larger number of people understand and follow the sport. The numerical indicators of the popularity of a sport are the number of spectators at sport events, its TV popularity ratings, and the number of sport clubs and athletes [27]. By all the criteria, wrestling in the capital is much more popular than in other Croatian cities.

The basic hypothesis is that there are differences in parental support between children of different ages, which is also logical from the perspective of the parent–child relationship in childhood or adolescence. Adolescence is a transitional phase to adulthood in which children seek as much autonomy as possible [28]. Another hypothesis is that there are differences in parental support in environments in which wrestling is popular and those in which it is not. The expectation is that some parents prefer to support their children when the sport is popular.

The aim of the research was to examine differences in parental support among wrestlers of different age categories, and between those coming from communities in which wrestling is a popular sport versus those in which it is less popular. The findings should provide new insights into the support of parents for young athletes.

## 2. Materials and Methods

### 2.1. Sample of Participants

The sample of participants consisted of 172 male wrestlers from 20 Croatian wrestling clubs. Several subsamples were created by the participants’ ages and whether they were residents of the capital or another Croatian city. We can say that the sample was representative because it consisted of about 60% of young wrestlers of this age from more than 90% of the wrestling clubs in the country. Seven female wrestlers filled in the questionnaire; however, they were not part of the sample because there were too few of them and they were not foreseen in the study project. The exclusion of girls from the study was due to the short tradition of female wrestling in Croatia [29]. The sample included young female wrestlers who had been wrestling for at least one year. Because wrestling training starts at the age of 10 years [30], the youngest wrestlers in the sample were 11 years old. In accordance with the International Wrestling Rules, the following age categories were considered: precompetitors (*n* = 53; age: 11.58 ± 0.50 years), boys U15 (*n* = 75; age: 14.00 ± 0.84 years), and cadets (*n* = 44; age: 16.29 ± 0.46 years). In the age category of boys U15, children of 14 and 15 years of age have the right to compete; however, this right may also be granted to children of 13 years subject to the permission of their parents. Wrestlers from 9 wrestling clubs from the capital city were represented in the sample (*n* = 93; age: 14.15 ± 1.83 years), whereas 11 clubs were from other Croatian cities (*n* = 79; age: 13.42 ± 1.85 years).

### 2.2. Sample of Variables

Participant responses to 25 items of the Parental Support Scale for Children in Sports [9] were the variables of our research. The questionnaire was originally designed in the Croatian language and was validated in samples of individual and team sports [9,14,31], as well as in samples of younger-aged athletes [31,32]. The questionnaire consisted of four subscales: 1. Parental beliefs in benefits of doing sports (9 items: 1, 3, 9, 11, 13, 16, 17, 19, and 25 (example: My parents think it is important for me to do sports to be healthier)); 2. Ensuring material conditions for doing sports (6 items: 6, 7, 14, 15, 22, and 23 (example: My parents pay for my extracurricular sports activities)); 3. Learning from role models (3 items: 4, 12, and 20 (example: My family and I often do a sporting activity together, regardless of our age differences)); 4. Positive reinforcement (7 items: 2, 5, 8, 10, 18, 21, and 24 (example: When talking about me to other people, my parents like to point out that I play sports)). Respondents were presented with 25 statements and were required to answer to what extent they agreed with them. Responses to the 25 items on the questionnaire were provided on a five-point Likert scale ranging from “strongly agree” to “strongly disagree.” The research used unpublished data from the doctoral dissertation Social Environment and Youth Participation in Wrestling [33]. The research was conducted in compliance with the Declaration of Helsinki and was approved by the Ethical Committee of the Faculty of Kinesiology, University of Zagreb (approval number: 57/2019).

### 2.3. Data Processing Methods

Data were processed by the program package Statistica for Windows, version 14.0 (TIBCO, Software Inc., Palo Alto, CA, USA). To test the reliability of the scale (questionnaire), Cronbach’s alpha and interitem correlation (IIC) were computed. All the variables were processed by descriptive statistics (median, mode, and mode frequency), whereas for the subscales, the mean and standard deviation were computed. The differences among the subscales were determined using the Friedman ANOVA and Wilcoxon matched-pairs test. For the determination of the differences among age groups, the Kruskal–Wallis test was used, and for the differences between the capital and other cities, the Mann–Whitney U test and Wilcoxon matched-pairs test were utilised. The level of statistical significance was set at *p* < 0.05.

### 2.4. Research Protocol

After the study was approved by the administrators of the wrestling clubs, parental consent was obtained. Test dates were then arranged with the clubs. The clubs were asked to provide a room where the children could complete the questionnaire at their leisure. The young wrestlers filled out the questionnaire anonymously and without the presence of a coach in a quiet place in the club rooms before or after training in groups of from five to ten. Because wrestlers of different age groups train at different times, the data were separately collected by age group. The children filled out the questionnaire with a pen on paper, as the youngest group required additional explanations that would not have been possible with the online method. It took approximately 15–30 min to complete the questionnaire. The method for completing the questionnaire was explained to all participants in the same way, and the research supervisor was available to clarify any ambiguities related to it. In each session, the principal investigator informed the children that they did not have to answer a question if it made them uncomfortable. Data collection took place at the beginning of the competition season. Data were collected within one month in all cities in which there were wrestling schools. Incorrectly completed questionnaires (many missing answers, the same answer for all questions) were removed from the sample (out of 196, 172 were correct).

## 3. Results

From Table 1, it is obvious that the reliability parameters (Cronbach’s alpha and IIC) indicate a high reliability of the measuring instrument in both the entire sample of participants and in each subsample.

Significant age differences (Table 2) were obtained in Subscale 1 (Parental beliefs in benefits of doing sports) and Subscale 4 (Positive reinforcement from parents). Moreover, a significant difference was obtained between the milieus (wrestling is popular vs. wrestling is less popular) in Subscale 4 (Positive reinforcement from parents).

Table 3 reveals statistically significant age differences in the responses to Items 8 (My parents are proud of me doing sports (*p* = 0.02)), 9 (My parents want me to do sports so that I would develop agility and strength (*p* = 0.02)), 16 (When they talk to other people about me, my parents are happy to point out that I do sports (*p* = 0.05)), and 18 (My parents are happy when my sport teammates meet up at our place (*p* ≤ 0.01)). Graphs of significantly different variables can be found in the Appendix A.

In Table 4, statistically significant differences are obvious between the wrestlers from the capital and those from other Croatian cities in the responses to Item 9 (My parents think athletes are a good company for me to keep (*p* = 0.04)) and Item 15 (My sports equipment was financed by my parents (*p* = 0.03)). Graphs of significantly different variables can be found in the Appendix A.

## 4. Discussion

The study was conducted with the aim of determining the differences in parental support between wrestlers of different age groups, and between those from communities in which wrestling is a popular sport and those in which it is less popular. This research demonstrated that age and popularity were related to parental support, which was determined as remarkably high. Comparable results (i.e., high parental support) were recorded in other studies as well [9,16]. Analysing the differences between the age groups, we found that there were significant differences in Subscale 1 (Parental beliefs in benefits of doing sports) and Subscale 4 (Positive reinforcement from parents). It is interesting that the group of boys U15 scored lower than both the younger (precompetitors) and older (cadets) wrestlers, which can be explained by their developmental stage. At this stage, boys commence their phase of specialisation; wrestling-specific contents are more focused on training, and first combat experiences are gathered. This is the stage at which children want more parental praise and understanding [34]. We can assume that, for the same reason, the boys U15 responses were lower on Subscale 1 as well (Parental beliefs in benefits of doing sports). First combat experiences can lead to fears and dilemmas for parents. The involvement of parents in their children’s sports depends not only on their gender (mother or father), sports experience, or lack of sports experience, but also on the child’s current stage of sports development [35]. Comments are very important for children at this stage in order for them to properly perceive their competence in sports, and their self-confidence depends on this. Children should not only receive comments from primary sources (coaches, teammates), but also from secondary sources (friends, parents) [36]. Comments are very important for children at this stage to properly assess their competence in sports, and their self-confidence depends on this. To support their children in sports, parents need to know something about the developmental stages in sports: “Before kids can play like a pro, they need to enjoy playing the game like a kid,” said Steve Locker, an international soccer player and coach [36]. Parental support should not turn into parental pressure, and the line is very thin. Although this is not an article about parental pressure, we must note this very thin line. Research showed that parents who are prone to pressure are willing to push their children towards early athletic specialisation, which is not good for either their mental or physical development [36,37,38]. Positive reinforcement from parents to children is higher in milieus in which wrestling is popular. The popularity of the sport had no effect on the other subscales. Parental comments on children’s sports depend, to a large extent, on whether the parents feel competent enough to comment on them [39]. In milieus in which wrestling is popular, more people practice it and/or follow wrestling competitions. In these communities, there are many more competent people who have no problem commenting on their children’s sport. Therefore, it was logical to expect that a lower level of positive support for children in sports would be present in the milieus in which wrestling is less popular. In Subscale 3 (Learning from role models/parents), no significant difference was found between the groups; however, we should emphasise here that the scores on this scale were low. Such a finding causes concern because the lack of positive role models was demonstrated to be one of the main reasons that children give up sports [40]. Similar findings were obtained by Crnjac in 2017. In the sample of 472 adolescent athletes from combat sports, the scores were between 4.06 and 4.48 for Subscales 1, 2, and 4, whereas the score for Subscale 3 was 3.57 [15]. Unfortunately, results on problematic Subscale 3 (Learning from role models/parents) cannot be compared with the findings of other studies because, apart from the Parent Support Scale Questionnaire [11], in which the Sports Habits of Parents subscale is included, no one has specifically analysed this problem, to the best of the authors’ knowledge. Sage (1980) conducted similar research and demonstrated that the variable Time spent participating with their offspring was lower and weakly related to socioeconomic status [41]. We can assume that more time spent with children means more opportunities to offer role model examples. All research conducted with the Parental Support Scale for Children in Sports questionnaire [9] points to the same problem: parents should demonstrate more by their own example. In our research as well, in all the groups, the same problem was ascertained as far as parents were concerned—a very high willingness to encourage children to participate in sports, a high willingness to finance children’s engagement in sports, and the attitude that sports are beneficial for their children, but not a very high willingness to show children parental attitudes towards sports by example. In this study, there was no age difference in Subscale 3 (Learning from role models/parents). This is interesting because it was expected that the oldest group would have a lower score on this scale. At their age, the adjustment to adulthood takes place [28]. This is the time when children run away from anything that they believe might hinder their transition to adulthood. The overinvolvement of parents in sports may also be a reason why young people give up sports [28].

If individual items of the questionnaire are analysed separately, then age differences occurred in 4 out of 25 items, whereas differences between the groups by wrestling popularity were recorded in two items. In Items 8 (My parents are proud of me doing sports), 9 (My parents want me to do sports so that I would develop agility and strength), and 16 (When they talk to other people about me, my parents are happy to point out that I do sports), the U15 wrestlers differed from the wrestlers of both the younger and older age groups. Children usually commence practicing wrestling at the age of 10 years [30,42]. The youngest group in our research had an average age of 11.5 years. The children should be at the stage of multilateral sports development and not yet at the specialisation stage; thus, they have not yet started seriously competing. Wrestlers of the second age group (U15) are participating in their first serious matches, whereas the cadets already gathered some competitive experience. From this aspect, the U15 wrestlers and their parents have the most dilemmas; children have their first combats but do not have any previous experience as to whether is it safe or how dangerous it is. Parents may have different perceptions of injuries and safety risks and encourage children to play sports [43]. Anxiety is a normal phenomenon in combat sports [44], but it is reduced with experience [44,45]. In Item 18 (My parents are happy when my sport teammates meet up at our place), the youngest group of wrestlers differed from the other two age groups. Our assumption is that that the reason for this finding is not the sport itself but the children’s age—at this age, they are too young to wonder across cities alone, someone needs to bring them to meet their friends, and they need special attention, which is not the case with the older age groups.

Similar to age differences, there were no differences between wrestlers coming from different milieus in terms of wrestling popularity in the majority of the questionnaire items except for two: My parents think athletes are a good company for me to keep and My sports equipment was financed by my parents. It is interesting that in the cities where wrestling is popular and widespread, parents have a dilemma as to whether colleagues from sports are good company for their children. We can assume that the dilemma arises from the fear of parents related to the violent behaviour of the young. It is obvious that parents have fears related to the safety of their children, and we can assume that this phenomenon is more prevalent in environments in which there are more wrestlers and wrestling clubs [43]. However, previous research on bullying demonstrated that the athletes from combat sports are not more often culprits of bullying or other aggressive behaviours than athletes from other sports [46]. In fact, most violent behaviour and aggression was committed by team sport athletes, whereas combat sport athletes have less often been victims compared with athletes from other sports in Croatia [46,47], where wrestling is most popular in its capital—Zagreb. Moreover, in the capital, citizens’ personal incomes are, on average, higher than in other parts of Croatia [48]. Therefore, it is feasible to assume that the higher personal incomes are the reason that the citizens of Zagreb are more willing to finance their children’s sports than their peers from other parts of Croatia.

## 5. Conclusions

This research was conducted with the aim of examining parental support for young wrestlers and to ascertain the possible relationships between this support, wrestlers’ ages, and the popularity of wrestling in the milieus from which they came. Parental support for young wrestlers was very high; on a scale from 1 to 5, the scores were in the range of from 4.3 to 4.5 for Subscales 1, 2, and 4, whereas Subscale 3 (Learning from role models/parents) was rated 3.75 on average. Parents believe that the sport is good for their children, they are ready to finance it, and they support their children in it; however, their willingness to show by example is lower. The analysis by age showed that when children enter the specialisation stage of their sports development, they seek more support from their parents, while parents have concerns about their children’s first combats. The popularity of the sport is related to parental support. In communities in which wrestling is more popular, parents better understand and know the sport, and so, they can participate in commenting on it and directly support their children. Moreover, our research showed that the parents in areas with higher personal incomes more willingly spend more money on their children’s sport, which is not necessarily related to the popularity of the sport.

Parents should become better acquainted with their children’s sports so that they can provide better support and comment on them together with their children, and they should lead by their own example. Coaches are the ones who could suggest these things to parents so that the parent–child bond in sports can grow stronger. However, caution is always recommended because there is a very thin line between parental support and parental pressure.

### Study Limitation

Several studies already dealt with this problem; however, the authors were mainly focused on the creation of new tools. Therefore, the results of these studies cannot be numerically compared but can only be commented on and compared in general because of the different methodologies used. Future studies should also include a sample of female wrestlers, whose numbers significantly increased and are sufficient for research. Our assumption is that parental support is not the same for males and females.

## Figures and Tables

**Table 1 jfmk-08-00065-t001:** Questionnaire reliability parameters (Cronbach’s alpha, interitem correlation (IIC)) for all participants and all groups separately.

	*N*	Cronbach’s Alpha	IIC
All groups	172	0.87	0.25
Precompetitors	53	0.81	0.19
Boys	79	0.83	0.21
Cadets	44	0.90	0.30
Capital city	93	0.87	0.25
Other cities	76	0.88	0.25

Note. *N*: number of respondents; IIC: interitem correlation.

**Table 2 jfmk-08-00065-t002:** Descriptive statistical parameters (mean and standard deviation) for all subscales and the differences among them (Friedman ANOVA, Wilcoxon matched-pairs test).

	Precompetitors	Boys	Cadets	Friedman ANOVA	Capital	Other Cities	Wilcoxon Matched Pairs Test
Subscales	Mean ± SD	Mean ± SD	Mean ± SD	(ANOVA Chi sqr.)	Mean ± SD	Mean ± SD	
Subscale 1	4.34 ± 1.01	4.16 ± 1.10	4.36 ± 0.95	Chi sqr. = 6.22	4.36 ± 0.99	4.24 ± 1.03	Z = 1.60
				*p* = 0.04			*p* = 0.11
Subscale 2	4.57 ± 0.79	4.53 ± 0.84	4.53 ± 0.82	Chi sqr. = 2.33	4.55 ± 0.78	4.53 ± 0.86	Z = −0.31
				*p* = 0.31			*p* = 0.75
Subscale 3	3.82 ± 1.13	3.56 ± 1.33	3.88 ± 1.25	Chi sqr. = 4.66	3.71 ± 1.19	3.74 ± 1.32	Z = −0.00
				*p* = 0.09			*p* = 1.00
Subscale 4	4.59 ± 0.72	4.36 ± 0.94	4.63 ± 0.63	Chi sqr. = 10.57	4.52 ± 0.80	4.48 ± 0.82	Z = 1.18
				*p* < 0.01			*p* = 0.02

Note. Subscale 1: Parental beliefs in benefits of doing sports; Subscale 2: Ensuring material conditions for doing sports; Subscale 3: Learning from role models; Subscale 4: Positive reinforcement from parents.

**Table 3 jfmk-08-00065-t003:** Descriptive statistical parameters (median, mode, mode frequency) for all three age groups and the differences among them (Kruskal–Wallis ANOVA).

	Precompetitors (*n* = 53)			U 15 (*n* = 75)			Cadets (*n* = 44)			Kruskal–Wallis ANOVA	
Item	Median	Mode	F Mode	Median	Mode	F Mode	Median	Mode	F Mode	H	*p*
1	5	5	47	5	5	58	5	5	34	4.35	0.11
2	5	5	45	5	5	56	5	5	34	1.42	0.49
3	4	4	17	3	3	32	4	5	17	3.78	0.15
4	4	3	18	3	3	21	4	5	17	1.09	0.58
5	5	5	38	5	5	47	5	5	26	5.46	0.07
6	5	5	49	5	5	59	5	5	34	1.09	0.58
7	5	5	43	5	5	63	5	5	37	7.42	0.02 *
8	5	5	48	5	5	60	5	5	39	7.42	0.02 *
9	5	5	46	5	5	52	5	5	33	4.24	0.12
10	5	5	43	5	5	56	5	5	38	0.42	0.81
11	4	5	23	4	5	37	4	5	21	1.77	0.41
12	4	5	23	4	5	33	5	5	26	4.00	0.10
13	5	5	30	4	5	32	5	5	25	0.02	0.99
14	5	5	38	5	5	53	5	5	29	5.95	0.75
15	5	5	40	5	5	57	5	5	30	5.95	0.05 *
16	5	5	39	5	5	48	5	5	35	5.81	0.06
17	5	5	38	5	5	43	5	5	31	12.27	<0.01 *
18	4	5	25	4	5	29	5	5	29	0.44	0.80
19	5	5	35	5	5	51	5	5	29	1.31	0.52
20	4	5	18	4	5	28	4	5	15	4.92	0.09
21	5	5	33	4	5	36	5	5	23	1.09	0.58
22	5	5	35	5	5	46	5	5	28	0.48	0.79
23	5	5	33	5	5	46	5	5	22	1.36	0.51
24	5	5	38	5	5	48	5	5	29	2.36	0.31
25	5	5	36	5	5	42	5	5	26	4.35	0.11

* Statistically significant difference between groups.

**Table 4 jfmk-08-00065-t004:** Descriptive statistical parameters (median, mode, mode frequency) and differences (Mann–Whitney U test) between wrestlers from the capital city and those from other cities.

	Capital City (*n* = 93)			Other Cities (*n* = 79)			Mann–Whitney U Test		
Item	Median	Mode	F Mode	Median	Mode	F Mode	U	Z	*p*-Value
1	5	5	79	5	5	60	3370.50	0.93	0.35
2	5	5	75	5	5	60	3485.00	0.44	0.66
3	3	3	38	4	5	27	3341.50	−1.02	0.31
4	4	Multiple	25	4	5	28	3493.00	−0.55	0.58
5	5	5	60	5	5	51	3625.00	0.00	1.00
6	5	5	76	5	5	66	3616.50	−0.17	0.86
7	5	5	80	5	5	63	3417.00	0.79	0.43
8	5	5	80	5	5	67	3629.00	0.14	0.89
9	5	5	79	5	5	52	3010.50	2.04	0.04 *
10	5	5	77	5	5	60	3374.00	0.92	0.36
11	5	5	47	4	5	34	3143.00	1.63	0.10
12	4	5	45	4	5	37	3459.50	0.66	0.51
13	5	5	54	4	5	33	3110.00	1.73	0.08
14	5	5	65	5	5	55	3669.00	0.01	0.99
15	5	5	77	5	5	50	2948.50	2.23	0.03 *
16	5	5	68	5	5	54	3505.00	0.52	0.61
17	5	5	61	5	5	51	3596.00	−0.24	0.81
18	5	5	51	4	5	32	3208.50	1.43	0.15
19	5	5	68	5	5	47	3118.50	1.70	0.09
20	4	5	29	4	5	32	3296.50	−1.16	0.25
21	5	5	53	4	5	39	3557.00	0.36	0.72
22	5	5	54	5	5	55	3090.00	−1.66	0.10
23	5	5	55	5	5	46	3612.50	−0.19	0.85
24	5	5	60	5	5	55	3455.00	−0.67	0.50
25	5	5	58	5	5	46	3564.00	0.33	0.74

* Statistically significant difference between groups.

## Data Availability

Data and other supplementary files can be found at: https://drive.google.com/drive/folders/1rtnk4GU_8aGZBVJyiqrOKG3zbMvUSD17?usp=share_link (accessed on 1 April 2023).

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
