# Peer review of "Effects of Age and Popularity of Sport on Differences among Wrestlers’ Parental Support: An Exploratory Study"

_jfmk, 2023, doi:10.3390/jfmk8020065_

Round 1

Reviewer 1 Report

I thank the authors for the opportunity to review this manuscript.

In the introduction, it is necessary to rely on the literature present on the topic and related to the research focus, not the tools used, which can be discussed in the method. I recommend expanding the introduction.

With respect to the aims, it is best to focus on the various hypotheses that are intended to be tested.

In the method section, it is unthinkable to analyze only one scale relative to the study, it is necessary to be able to make comparisons, correlations with additional other scales in order to arrive at significant conclusions.

The study might be interesting, but as formulated it is fallacious.

I recommend evaluating some means of concordance or discordance and doing further analysis with other variables. These are not generalizable results, either because of the number of participants or the method used. The results obtained could be related to other variables not analyzed.

Best regards

Reviewer 2 Report

Dear Editor,

In this study the authors examine differences in parental support among wrestlers of different age categories and between those coming from the communities where wrestling was a popular sport versus the communities in which it was less popular.

Although the study has the potentiality of being shared with the scientific community, I believe that the manuscript would benefit from a major revision with the attempt to better support their experimental setting.

Abstract should start with a paragraph dedicated to a brief description of the background.

Introduction:

The theoretical framework is scarce, they should clearly describe the scientific evidence that supports the hypothesis they have raised.

Method:

A lot of necessary information is missing in methods section:

-        Experimental procedures should be better defined.

-        More information should be provided about the participants’ characteristics.

-        The intervention protocol should be better described.

-        Anthropometric measurements tests presuppose a protocol.

This element is missing from the methodological description, which may imply an impossibility of replicating the study due to a lack of clarity in this regard.

1.               The Discussion should be enriched with the existing theory. The authors should clearly describe the scientific evidence that supports their findings. In addition, they should start with a first paragraph describing the main aims and then the main results.

Kind regards

Round 2

Reviewer 1 Report

Dear authors,

certainly the manuscript has improved, but basic critical issues about the sample number and instruments used remain. My only advice is to tone down the title a bit by calling it "an exploratory study" given the profound limitations.

Best Regards

Author Response

Dear Reviewer,
Thank you for the suggestions, which have certainly improved the study. Regarding the last suggestion, we have added "an exploratory study" to the title of the study.

So the full title now reads:

Effects of Age and Popularity of Sport on Differences among Wrestlers' Parental Support: An Exploratory Study

Best regards,

Hrvoje

Reviewer 2 Report

No further comments 

Author Response

Thanks for the review, you helped improve the paper.

Best regards,

Hrvoje

Round 3

Reviewer 1 Report

Thank you